# A firm-level analysis of Chinese commercial health insurance surrender

Ling Tian[1,2‡], Haisong Dong[1‡]*

**1** School of Economics and Management, Wuhan University, Wuhan, China, **2** National Institute of Insurance Development, Wuhan University, Ningbo, China

‡ LT and HD contributed equally to this work and should be considered co-first authors.
* hsongdong@163.com

**Data Availability Statement:** All relevant data are within the manuscript and its Supporting information files.

**Funding:** The author(s) received no specific funding for this work.

## Abstract

Based on the unbalanced panel data of Chinese professional health insurance companies from 2011 to 2021, the paper constructs "PW+PCSE" model to empirically investigate the main factors affecting the commercial health insurance surrender in China from the company level. The results show that asset-liability ratio has a significant positive effect on health insurance surrender rate. The value preservation and appreciation rate of capital and R&D expenditure rate both have significant negative effects on health insurance surrender rate. These studies bring important enlightenment for domestic health insurance companies to avoid surrender risk.

## 1. Introduction

Since the reform and opening-up for more than 40 years, China's commercial health insurance industry develops rapidly [1–5], at the same time, the surrender amount of the commercial health insurance industry is also high, and the surrender problem is relatively serious [6, 7]. In 2019 and 2020, for example, the surrender amount of several professional health insurance companies increased: the surrender amount of "Ping An Health" was CNY 2.0321 million and CNY 3.1678 million respectively, with a year-on-year growth of 55.89%. The surrender amount of "Ruihua Health" was CNY 0.0285 million and CNY 2.9531 million respectively, which increased 102.62 times year-on-year. The surrender amount of "Kunlun Health" climbed from CNY 79.4485 million to CNY 158.5284 million, an increase of 99.54%.

Surrender means that when the insurance contract is not fully performed, the applicant applies to the insured, the insurer agrees to terminate the legal relationship between the two parties determined by the contract, and the insurer returns the cash value of the policy in accordance with the Insurance Law and the contract. The high surrender rate will directly affect the normal operation of commercial health insurance companies, damage the image of commercial health insurance companies, and is not conducive to the sustainable development of commercial health insurance industry. Based on this, this paper takes the professional health insurance companies in China as the research object (Although there are 164 insurance companies operating health insurance business in China in 2020, due to the diversified insurance business, the surrender amount of commercial health insurance and the breakdown of the

**Competing interests:** The authors have declared that no competing interests exist.

financial characteristics of companies operating health insurance business are not disclosed in detail in the annual report, so the research object is targeted at professional health insurance companies), aiming to empirically investigate the main factors affecting the surrender of commercial health insurance in China from the company level, so as to verify the "policy replacement hypothesis" and provide useful value information for the stable development of commercial health insurance.

The original idea of the "Policy substitution hypothesis" was first proposed by Outreville in 1990, and the surrender behavior hypothesis was formally advocated by Russell in 2013. This hypothesis holds that in addition to the external factors in the long-term insurance market, the phenomenon of "poaching" and "grabbing orders" among insurance companies caused by internal competition in the long-term insurance market, as well as the fact that the original policy may lose its advantage due to the changes in the operating conditions of insurance companies in the long-term insurance market, is one of the important factors affecting the surrender behavior. It is believed that the motivation of the policy owner to surrender the policy is that the insurance degree of the original policy is worried and the new policy is favored by the original policy holder in terms of yield, insurance or product rate, so the original policy holder is willing to bear the loss of surrender and chooses to buy the new policy. That is, company-level factor indicators——company operating status [6, 8–11] and new policy business [12–14] are the main influencing factors of long-term insurance surrender.

The asset-liability ratio, as one of the solvency indicators of a company, reflects the degree of protection of the company's assets to creditors' rights and interests. The higher the asset-liability ratio, the weaker the company's long-term solvency and the worsening of the company's operating condition. Policyholders will opt to surrender the insurance out of concern about the insurance guarantee prospect, which may lead to an increase in the surrender rate [6]. R&D expenditure rate, as one of the company's stable operation indicators, reflects the intensity of the company's investment in product development. The higher the R&D expenditure rate, the greater the company's R&D and technology investment, the stronger the product innovation consciousness, the willingness to design more products to meet the needs of target customers, the stronger the company's core development ability, and the more opportunities for policy holders to choose suitable policies. Therefore, the surrender rate may be reduced. As one of the indicators of a company's development ability, the value preservation and appreciation rate of capital reflects the operating efficiency and safety status of the company's capital. The higher the value preservation and appreciation rate of capital, the stronger the company's ability to use the capital invested by investors to create profits, the better the company's capital preservation status, the faster the growth of owners' equity, and the corresponding protection of creditors' debts. The stronger the company's growth, the policy holders have no worries about the personal protection provided by the policy, so the surrender rate may decrease. Based on the above analysis, this paper proposes the following hypotheses:

**Hypothesis 1**. *The asset-liability ratio has a significant positive effect on the surrender rate of commercial health insurance.*

**Hypothesis 2**. *Both R&D expenditure rate and the value preservation and appreciation rate of capital have a significant negative effect on the surrender rate of commercial health insurance.*

## 2. Review of relevant literature

At present, the surrender of life insurance (life insurance here is a broad concept, including commercial health insurance, so these literatures are equivalent to the study of commercial

health insurance surrender) is a study on the verification of four classical surrender hypotheses.

In terms of the "emergency fund hypothesis", Outreville (1990) [15] analyzed the deniability of the life insurance market in the United States and Canada during 1955–1979 with data provided by the American Council of Insurers and the International Life Insurance Marketing Research Association, built a multiple linear regression model and found that the unemployment rate had a significant positive effect on the deniability rate. Kuo et al. (2003) [16] collected the data of policy surrender rate of the United States from 1951 to 1998, and built VAR model based on it, investigated the dynamic relationship between unemployment rate, 90-day Treasury bond interest rate and policy surrender rate, and concluded that in the short term, unemployment rate had a more significant impact on policy surrender rate than interest rate. In the long run, interest rate had a more prominent effect on surrender rates than unemployment rate. Kim (2005) [17] took the relationship between the surrender rate and interest rate, policy implementation time, unemployment rate, economic growth rate and seasonal effect as the research purpose, and used the Logit function and complementary Log-Log function successively to model the surrender rate. The results showed that: The Logit model and the complementary Log-Log model are significantly better than the existing Arc-tangent curve model. Unemployment rate, interest rate, economic growth rate, seasonal effect and policy implementation time had highly significant effects on the surrender rate, and the unemployment rate had a significant positive impact on the surrender rate. Jiang (2010) [18] constructed ECM to avoid the problem of white noise in traditional multiple co-integration vectors, and studied the relationship between multiple co-integration vectors and surrender rate. Empirical results showed that "emergency fund theory" and "interest rate replacement theory" were valid in both short and long term, that is, unemployment rate and interest rate had a significant impact on surrender rate. Barucci et al. (2020) [19] investigated the driving factors for the surrender of life insurance contracts of a large Italian insurance company, and found that personal financial/economic difficulties had a significant positive impact on the surrender rate. Cole and Fier (2021) [20] concluded that loan activity was an important factor affecting life insurance surrender. Shim et al. (2021) [21] selected panel survey data of Korean Retirement Income Study (KReIS) with many diverse dimensions to determine which variables had a decisive effect on the lapse and applied the lasso regularized regression model to analyze it empirically and used random forest interpolation to compensate for the missing values. According to the study: (1) In terms of the household variables, the non-existence of old dependents, the existence of young dependents, and employed family members increased the surrender rate. (2) In terms of the individual variables, divorce, non-urban residential areas, apartment type of housing, non-ownership of homes, and bad relationship with siblings increased the surrender rate. (3) In terms of the financial variables, low income, low expenditure, the existence of children that incurred child care expenditure, not expecting to bequest from spouse, not holding public health insurance, and expecting to benefit from a retirement pension increased the surrender rate.

In terms of "interest rate substitution hypothesis", Pesando (1974) [22] mainly analyzed from the perspective of "interest rate substitution theory" and believed that interest rate and surrender rate were significantly positively correlated. The reason was that the rising expected market interest rate increased the expected interest rate of future new products, which decreased the price of new policies and increases the tendency of policy holders to surrender. Babbel (1995) [23] further studied the impact of interest rate changes on the surrender rate of life insurance companies by analyzing the impact of interest rate changes on the cash flow of life insurance companies. After measuring the impact of inflation rate on the cost of life insurance, a conclusion was drawn: the change of interest rate would promote the surrender rate of

policy holders, that is, the interest rate had a significant and positive correlation with the sur-render rate. Tsai et al. (2002) [24] took the American life insurance market as the sample object, built the VAR model based on the collected surrender rate data, and analyzed the dynamic relationship between interest rate and surrender rate. The research found that there was a significant long-term influence relationship between the two, and the two were highly positively correlated. Knoller et al. (2016) [12] believed that interest rate was an important fac-tor affecting the surrender of traditional life insurance through theoretical analysis. Wei et al. (2019) [8], through qualitative analysis, believed that interest rate had significant influence on life insurance surrender rate, which supported the "interest rate substitution theory".

In terms of the "payment depreciation hypothesis", Babbel (1979) [25] established a model to measure the cost of life insurance, used the net cost-benefit ratio to investigate the impact of inflation rate on the cost of life insurance, and found that: If the premiums and claims of life insurance policies could not be fully adjusted and the policy holders did not have the illusion of money, then the life insurance surrender rate was bound to rise, that is, the inflation rate had a significant positive impact on the surrender rate. Babbel (1981) [26] took the Brazilian life insurance market as the research object and discussed the impact of indexed life insurance policies on the sales of life insurance products under the condition of expected inflation. The research found that: in the period of inflation, indexed life insurance policies failed to solve the problem of high price of life insurance products. The inflation rate would reduce the sales of life insurance products, which indirectly indicates that the inflation rate had a significant and positive correlation with the life insurance surrender rate. Wei et al. (2019) [8], through quali-tative analysis, believed that the inflation rate had a significant impact on the life insurance sur-render rate, which supported the "payment depreciation theory".

In terms of "policy replacement hypothesis", Mauer and Holden (2007) [9] used public data of the American life insurance market to analyze the impact of corporate financial pressure, product rates, product structure ratio, company size and other factors on the surrender rate of life insurance products. Kiesenbauer (2012) [10] adopted the Logit model to analyze the mac-roeconomic indicators and operating conditions of 133 German life insurance companies from 1997 to 2009. The study found that: interest rate and contingency fund assumptions only applied to unit-linked operations. The business condition of the company was an important factor affecting the surrender rate of life insurance. Russell et al. (2013) [13], based on the panel data of 51 states in the United States from 1995 to 2009, conducted an empirical test on the "policy substitution" effect affecting the surrender rate in the American life insurance mar-ket and found that the ratio of new policies (the proportion of new policies in the total policy premium income) had a significant impact on the surrender rate. Zhan and Chen (2013) [11] constructed a nonlinear Panel Smooth Transition Regression (PSTR) model using non-equi-librium panel data of Chinese life insurance market from 2001 to 2010, and came to the follow-ing conclusions: The total assets had a significant positive effect on the surrender rate. The company's establishment years and the average premium income both had a significant nega-tive impact on the surrender rate. Eling and Kiesenbauer (2014) [14], taking a German life insurance company as the research object, believed that product type or contract age and other product characteristics were important factors affecting the surrender rate. Knoller et al. (2016) [12] made an empirical analysis on surrender behavior of variable annuity contract by using Japanese individual policy data and found that premium income was an important factor affecting surrender of variable annuity contract. Yu et al. (2019) [6] used enterprise-provincial panel data of China's life insurance industry from 2005 to 2013 to study the determinants of the surrender rate of China's life insurance industry. It was concluded that the degree of busi-ness concentration and the years of establishment of the insurance company had a significant negative impact on the surrender rate. Company size and asset-liability ratio had significant

positive impact on the surrender rate. High surrender rates undermined insurers' financial soundness and hurt new business. Wei et al. (2019) [8] studied the driving factors of life insurance surrender risk in China under the economic crisis, and established the economization model of influencing factors of life insurance surrender rate at the company level. The results showed that the economic situation of insurance companies under the economic crisis was the main factor affecting the life insurance surrender rate.

From the existing research, the research literature on the surrender of commercial health insurance has been relatively rich. The academic contribution and value of this paper are as follows: Firstly, on the basis of previous studies, we continue to try to explore the main factors affecting the surrender of commercial health insurance in China through the empirical study at the company level, so as to verify the "policy replacement hypothesis", which is an academic exploration of great research significance. Secondly, according to the main conclusions, it provides valuable clues for health insurance companies to prevent surrender risks, insurance regulatory departments to formulate corresponding policies and promote the market-oriented reform of commercial health insurance product rates.

## 3. Study design

### 3.1 Variable selection and calculation

**3.1.1 Explained variables.** Commercial health insurance surrender rate: As for the calculation method of surrender rate, the existing literature generally adopts three methods: Method 1 is the surrender fee of this year divided by the premium income of this year; Method 2 is the surrender fee of this year divided by the total payout of this year; Method 3 is the surrender fee of this year/(long-term insurance liability reserve at the end of last year + premium income at the end of this year). The statistical caliber of numerator and denominator of the first two methods is inconsistent, which tends to underestimate or overestimate the surrender rate of the current year [11], resulting in the lack of rationality of the calculation results. Method 3 is the calculation method defined by the former China Insurance Regulatory Commission in the document "Standard for Statistical Analysis Index System of Insurance Companies". This method has the same statistical caliber and can get the surrender rate more accurately. Based on this, this paper adopts the third method to calculate the surrender rate of commercial health insurance. It is used as a quantitative index of commercial health insurance policy surrender in a certain period of time and as an explained variable.

**3.1.2 Explanatory variables.** Asset-liability ratio: calculated and expressed by dividing the total liabilities by the total assets. R&D expenditure rate: calculated and expressed by dividing R&D expenditure by total operating revenue. Value and appreciation rate of capital: calculated and expressed by dividing owners' equity at the end of the current year by owners' equity at the end of the previous year. Foreign-funded group company: Used to indicate whether foreign-funded group company (yes: its value is 1, no: its value is 0), is a dummy variable.

The names, units of measurement, symbols and definitions of the variables are described in Table 1.

### 3.2 Data sources

This paper takes "Ping An Health" (PA), "PICC Health" (RB), "Pacific Health" (TPY), "Hexie Health" (HX), "Kunlun Health" (KL), "Ruihua Health" (RH) and "Fuxing United Health" (FXLH) as the research objects. Based on the data of these seven professional health insurance companies from 2011 to 2021, the issue of commercial health insurance surrender is analyzed from the company level. Due to the late establishment of some professional health insurance companies or the lack of public disclosure of information and other reasons, some indicator

**Table 1. Definition description of each variable.**

| Variable name | Variable symbol | Variable description |
|---|---|---|
| Health insurance surrender rate (%) | SUR | Surrender fund at the end of the current year/(long-term health insurance liability reserve at the end of the previous year + premium income at the end of the current year) ×100 |
| Asset-liability ratio (%) | LEV | Total liabilities/total assets×100 |
| R&D expenditure rate (%) | RDE | R&D expenses/Total revenue×100 |
| Value and appreciation rate of capital (%) | CRNA | Owners' equity at the end of the current year/owners' equity at the end of the previous year×100 |
| Foreign-funded group company | F_group | Dummy variable;Foreign-funded group company or not (Yes = 1, no = 0) |

data are missing in some years. However, in order to preserve the information contained in the existing data as much as possible and improve the efficiency of estimation, the data used in this paper are non-balanced panel data. And because of small n and large T (n = 7, T = 11), it is a non-balanced long panel. Data for all variables are obtained from annual reports of professional health insurance companies. The data preprocessing software is "Excel" and the econometric analysis software is "Stata". In order to minimize the influence of extreme values on the research results, all data of continuous variables are winsorized based on 15% and 85% quantiles. At the same time, in order to make up for the missing data, the linear interpolation method is used to supplement the missing data.

## 3.3 Model expression

Generally speaking, the surrender rate of a health insurance company this year is not affected by the surrender rate of the previous year, that is, $SUR_t$ is not affected by $SUR_{t-1}$. Therefore, this paper considers the static non-equilibrium long panel data model. Chen (2014) [27] pointed out that in the long panel, due to small n and large T, individual dummy variables can be added to the possible individual fixed effects, and time trend items can be added to control the possible time fixed effects. Therefore, the model expression in this paper is set as:

$$SUR_{it} = X'_{it}\beta + rTime + u_i + \varepsilon_{it}(i = 1, 2, \cdots, 7; t = 1, 2, \cdots, 11)$$

Where, $X_{it=}$ (LEV, RDE, CRNA)', is a vector composed of explanatory variables varying with individual and time changes. Time is the time trend item, which is used to control the time effect. $u_i$ is individual effect. $\varepsilon_{it}$ is a random perturbation term that varies with individual and time, and it is independently and equally distributed. $cov(\varepsilon_{it}, u_i) = 0$, representing the difference in intercept.

## 4. Empirical analysis

### 4.1 Multicollinearity test

Before the empirical analysis, the multicollinearity test of independent variables [28, 29] is first conducted, as shown in Table 2. As shown in Table 2, the maximum variance enlargement factor (VIF) of all independent variables is 2.39, far less than 10, indicating that there is no serious multicollinearity problem in the model, and subsequent empirical analysis can be conducted.

**Table 2. Multicollinearity test results of independent variables.**

| Variable | VIF value | 1/VIF value | Result |
|---|---|---|---|
| LEV | 2.39 | 0.419 | There is no severe multicollinearity |
| RDE | 1.59 | 0.627 | There is no severe multicollinearity |
| CRNA | 1.33 | 0.754 | There is no severe multicollinearity |
| Time | 1.37 | 0.731 | There is no severe multicollinearity |
| company | | | |
| 2 | 2.01 | 0.497 | There is no severe multicollinearity |
| 3 | 2.01 | 0.497 | There is no severe multicollinearity |
| 4 | 1.87 | 0.536 | There is no severe multicollinearity |
| 5 | 2.00 | 0.499 | There is no severe multicollinearity |
| 6 | 2.20 | 0.455 | There is no severe multicollinearity |
| 7 | 1.82 | 0.549 | There is no severe multicollinearity |

## 4.2 Stability test and descriptive statistics of variable data

This paper uses unbalanced panel data, so it is necessary to check the stationarity of panel data to avoid the phenomenon of pseudo regression. The stability test method of variable data used in this paper is Fisher-ADF unit root test [30–33], and the test results of each variable are shown in Table 3. It can be seen from Table 3 that variables LEV, RDE, CRNA and SUR all pass the 1% significance level test, indicating that the panel data are stationary data, which ensures the accuracy of subsequent empirical analysis. The descriptive statistics of the above stationary variables are shown in Table 4.

## 4.3 Inter-group heteroscedasticity test, intra-group autocorrelation test and inter-group coincident correlation test

Since long panel data is used in this paper, the random disturbance item $\varepsilon_{it}$ may have inter-group heteroscedasticity, intra-group autocorrelation or inter-group covariance, so it needs to be tested. Table 5 shows the test results. Wald test shows that the P value of $\varepsilon_{it}$ is 0.088, which is significant at the level of 10%, with heteroscedasticity between groups. Wooldridge test shows that $\varepsilon_{it}$ has heteroscedasticity between groups. Pesaran test shows that there is no inter-group coincident correlation for $\varepsilon_{it}$, that is, there are inter-group heteroscedasticity and intra-group autocorrelation problems in this model, which need to be dealt with to ensure the reliability of empirical research results of the model.

**Table 3. Stability test results of each variable data.**

| Variable | Fisher-ADF value | P-value | Result |
|---|---|---|---|
| LEV | -6.241 | 0.0000*** | Stable |
| RDE | -4.378 | 0.0001*** | Stable |
| CRNA | -5.164 | 0.0000*** | Stable |
| SUR | -8.519 | 0.0000*** | Stable |

Note:

*** is significant at the level of 1%.

**Table 4. Descriptive statistics of each variable.**

|  | LEV | RDE | CRNA | SUR |
|---|---|---|---|---|
| Obs | 60 | 60 | 60 | 60 |
| Mean | 77.521 | 27.049 | 134.827 | 2.959 |
| Std. Dev. | 21.447 | 45.335 | 82.731 | 7.686 |
| Max | 99.74 | 353.56 | 501.81 | 38.615 |
| Min | 4.35 | 4.31 | 47.86 | 0.000 |
| Median | 84.065 | 15.445 | 111.91 | 1.089 |
| Skewness | -2.182 | 6.384 | 2.706 | 3.979 |
| Kurtosis | 7.052 | 46.409 | 11.389 | 17.653 |

**Table 5. Test results of inter-group heteroscedasticity, intra-group autocorrelation and inter-group coincident correlation.**

| Model test type | Checks statistical value | P-value | Result |
|---|---|---|---|
| Inter-group heteroscedasticity test | Wald: 12.41 | 0.088* | There is inter-group heteroscedasticity |
| Intra-group autocorrelation test | Wooldridge: 9.395 | 0.022** | There is intra-group autocorrelation |
| Inter-group coincident correlation test | Pesaran: -0.501 | 1.384 | There is no inter-group coincident correlation |

Note:

** and * are significant at the level of 5% and 10% respectively.

## 4.4 Analysis of the empirical results

Table 6: Column (1) is the least square dummy variable method (LSDV) used to estimate the model, and there is no inter-group heteroscedasticity, intra-group autocorrelation or inter-group coincident correlation in the corresponding model. Table 6: Column (2) uses panel correction standard error (PCSE) to estimate the model, and there is inter-group heteroscedasticity or inter-group coincident correlation, but no intra-group autocorrelation in the corresponding model. Table 6: Column (3) is used to estimate the model by "Prais-Winsten" estimation method (i.e."PW+PCSE"), and the corresponding model has inter-group heteroscedasticity or inter-group coincident correlation, and intra-group autocorrelation (assuming the random disturbance term $\varepsilon_{it}$ follows AR (1) process, and the autoregressive coefficients of each professional health insurance company are the same). Comparing column (1) and column (2), it can be found that the estimation coefficients of the two estimation methods are exactly the same, but the significance of the coefficients is different. Therefore, it can be shown that if the model has inter-group heteroscedasticity or inter-group coincident correlation, but there is no intra-group autocorrelation problem, the estimation coefficients of each explanatory variable will not be changed, but the standard error will be affected. By comparing columns (3) with columns (1) and (2), it is found that the estimated coefficients of "PW+PCSE" are different from those of columns (1) and (2), which indicates that if the model has the problem of inter-group heteroscedasticity or inter-group coincident correlation, and intra-group autocorrelation, it will not only change the estimated coefficients of each explanatory variable, but also affect the standard error.

Based on the problems of inter-group heteroscedasticity and intra-group autocorrelation in the model presented in this paper, and combined with the analysis in the previous paragraph, we can see in Table 6: Among the three estimation results in columns (1)—(3), "PW+PCSE (PCSE_AR1)" is the most robust estimation result. Therefore, PCSE_AR1 is taken as the benchmark regression result of this paper, and the empirical results are mainly analyzed. From

**Table 6. Model estimation results.**

| Variable | (1) | (2) | (3) |
|---|---|---|---|
| | **LSDV** | **PCSE** | **PCSE_AR1** |
| LEV | 0.0343* (0.0152) | 0.0343*** (0.0116) | 0.0336*** (0.0108) |
| RDE | -0.0274* (0.0138) | -0.0274*** (0.00666) | -0.0229*** (0.00656) |
| CRNA | -0.00337 (0.00259) | -0.00337* (0.00176) | -0.00292* (0.00172) |
| Time | -0.0702* (0.0303) | -0.0702*** (0.0166) | -0.0641*** (0.0169) |
| company | | | |
| 2 | -1.270*** (0.129) | -1.270*** (0.221) | -1.237*** (0.262) |
| 3 | -0.267 (0.176) | -0.267 (0.323) | -0.261 (0.414) |
| 4 | -1.668*** (0.0788) | -1.668*** (0.232) | -1.596*** (0.272) |
| 5 | -1.715*** (0.170) | -1.715*** (0.256) | -1.639*** (0.368) |
| 6 | -1.294*** (0.224) | -1.294*** (0.333) | -1.234*** (0.344) |
| 7 | -0.650* (0.289) | -0.650 (0.463) | -0.617 (0.558) |
| _cons | 0.694 (1.341) | 0.694 (0.938) | 0.518 (0.949) |
| Obs | 60 | 60 | 60 |
| R-squared | 0.769 | 0.769 | 0.668 |

Note:

*** and * are significant at the level of 1% and 10% respectively.

Table 6: Column (3), it can be seen that the estimated coefficient of asset-liability ratio (LEV) is 0.0336 (indicating that the surrender rate will increase by 0.0336 for each additional LEV unit), which passes the significance level test of 1%, that is, asset-liability ratio (LEV) has a significant positive effect on the surrender rate, thus Hypothesis 1 can be verified, this is consistent with the conclusion of Yu et al. (2019) [6]. The estimated coefficient of R&D expenditure rate (RDE) is -0.0229 (indicating that the surrender rate will decrease by 0.0229 for each additional RDE unit), which passes the significance level test of 1%, that is, R&D expenditure rate (RDE) has a significant negative effect on the surrender rate. The estimated coefficient of value and appreciation rate of capital (CRNA) is -0.00292 (indicating that the surrender rate will decrease by 0.00292 for each additional CRNA unit), which passes the significance level test of 10%, that is, value and appreciation rate of capital (CRNA) has a significant negative effect on the surrender rate, thus Hypothesis 2 can be verified.

## 5. Robustness test

In order to ensure the reliability of the empirical conclusions of the above benchmark regression model, the corresponding robustness test is conducted in this paper. The robustness test results are shown in Table 7, where column (1) is the regression of "PW+PCSE (PCSE_AR1)", so as to compare with the empirical results of the robustness test method. Specifically, two methods are used to verify the stability of the benchmark regression conclusions. Method 1, method of adding explanatory variables, that is, the dummy variable of foreign-funded group companies (F_group) is added into explanatory variables of the model, and a set of estimated results were obtained as shown in column (2). By comparing columns (1) and (2) in Table 7, it can be found that the empirical conclusions of the two are basically consistent, indicating that the baseline empirical estimation results are robust. Method 2, method of deleting explanatory variables, that is, deleting the value and appreciation rate of capital (CRNA) in explanatory variables of the model. Another set of estimated results is shown in column (3). By comparing columns (1) and (3) in Table 7, it can be found that, compared with the regression of "PW+PCSE

**Table 7. Model robustness tests.**

| Variable | (1) | (2) | (3) |
|---|---|---|---|
| | PCSE_AR1 | PCSE_1 | PCSE_2 |
| LEV | 0.0336*** (0.0108) | 0.0336*** (0.0108) | 0.0331*** (0.0104) |
| RDE | -0.0229*** (0.00656) | -0.0229*** (0.00656) | -0.0240*** (0.00607) |
| CRNA | -0.00292* (0.00172) | -0.00292* (0.00172) | |
| Time | -0.0641*** (0.0169) | -0.0641*** (0.0169) | -0.0572*** (0.0160) |
| F_group | | (omitted) | |
| company | | | |
| 2 | -1.237*** (0.262) | -1.237*** (0.262) | -1.220*** (0.244) |
| 3 | -0.261 (0.414) | -0.261 (0.414) | -0.290 (0.436) |
| 4 | -1.596*** (0.272) | -1.335*** (0.260) | -1.681*** (0.266) |
| 5 | -1.639*** (0.368) | -1.639*** (0.368) | -1.636*** (0.310) |
| 6 | -1.234*** (0.344) | -0.973*** (0.366) | -1.256*** (0.341) |
| 7 | -0.617 (0.558) | -0.356 (0.479) | -0.533 (0.540) |
| _cons | 0.518 (0.949) | 0.257 (0.974) | 0.196 (0.906) |
| Obs | 60 | 60 | 60 |
| R-squared | 0.668 | 0.668 | 0.659 |

Note:

*** and * are significant at the level of 1% and 10% respectively.

(PCSE_AR1)", the estimation coefficient, influence direction and significance level of each explanatory variable have not changed in general, that is, the empirical results of the benchmark are still robust. To sum up, the benchmark empirical conclusions of this paper have good stability.

# 6. Endogenous description

Generally speaking, the better the operating condition of the insurance company, the smoother the development of the insurance company, the smaller the chance of surrender; At the same time, the lower the surrender rate of the insurance company, it will also have a positive impact on the operating status of the insurance company, which is conducive to improving the financial status of the insurance company. Therefore, it is easy to have endogeneity problems in the selection of proxy variables for corporate financial characteristics. In the process of explaining variable selection, this paper does not use the absolute value of variables such as total assets, total liabilities, total owners' equity, R&D expenses and total operating revenue, but adopts the relative size of different variables. For example, the increase in the total liabilities of a health insurance company does not necessarily lead to the increase in the ratio of the total liabilities of the health insurance company to the total assets (i.e., asset-liability ratio: LEV), thus avoiding the possible endogenous problems in the model [34].

# 7. Conclusions, implications, and prospects

## 7.1 Conclusions

Based on the non-equilibrium long panel data of Chinese professional health insurance companies from 2011 to 2021, this paper constructs a "PW+PCSE" model, and empirically examines the main factors affecting the surrender of Chinese commercial health insurance from the company level. The main conclusions of this paper are as follows: firstly, there is a significant

positive correlation between LEV and SUR. Secondly, both the rate of R&D expenditure (RDE) and the rate of value and appreciation of capital (CRNA) have a significant negative effect on the surrender rate of health insurance (SUR).

## 7.2 Policy enlightenments

Based on the above main research conclusions, the policy enlightenments of this paper are as follows:

1. Strengthen asset and liability management of health insurance companies and control asset-liability ratio. Assets and liabilities management is the unified and coordinated management of assets and liabilities by health insurance companies according to the changes of economic environment and their own business conditions. Due to the significant positive correlation between the asset-liability ratio and the surrender rate, Chinese health insurance companies should pay attention to strengthen the management of assets and liabilities, reasonably arrange the insurance asset portfolio according to the characteristics of the insurance liability portfolio, and control and reduce the asset-liability ratio [35]. By matching the cost and benefit, maturity, nature and scale of assets and liabilities, we can guarantee the virtuous cycle of funds and the solvency of commercial health insurance companies and realize the coincidence of assets and liabilities cash flow.

2. Establish a sound research and development system for health insurance products, and stabilize the R&D expenditure rate. Due to the significant negative correlation between R&D expenditure rate and surrender rate, Chinese health insurance companies should actively establish a perfect system of commercial health insurance product development, and steadily increase the rate of R&D expenditure. A perfect commercial health insurance product research and development system should be a gradual and cyclic process, that is, market research, planning and demonstration, product design, training and promotion, exhibition sales, information feedback, supervision and assessment. Design and develop products according to the full and detailed market survey results, and improve and perfect the products through the feedback mechanism after the sale of new products, which can effectively avoid the problem of homogeneity and singleness of products, but also can keep the health insurance company's product research and development ideas active, and achieve product innovation. Health insurance companies should set up special funds for product research and development, determine the product positioning and scope suitable for their own characteristics, implement product differentiation strategy, design flexible and diversified products to meet the actual needs of target customers and reduce the occurrence of surrender of insurance.

3. Strengthen the asset management level of health insurance companies and raise the value and appreciation rate of capital. Since there is a significant negative correlation between the value-added rate of capital preservation and the surrender rate, Chinese health insurance companies should optimize the asset portfolio of insurance companies and strive to improve asset returns to realize capital preservation and appreciation. At present, the capital investment of Chinese health insurance companies is still excessively concentrated in bank deposits and bonds, and there is a risk of concentrated allocation. With the promulgation of the Interim Measures on the Management of the Use of Insurance Funds (revised version), the investment of insurance funds can be extended to unlisted equity, real estate, mortgage loans and other new assets, so as to expand the space of insurance asset allocation. In addition, the investment effect of the assets of health insurance companies is closely related to the choice of investment organization mode. Investment

organization mode is an effective means to ensure investment returns and prevent investment risks. Health insurance companies should adopt more specialized investment organization mode to further improve the value and appreciation rate of capital.

### 7.3 Research prospects

In the study of the impact of company-level characteristics on commercial health insurance surrender, this paper examines the direction, size and significance of the effect on commercial health insurance surrender from three perspectives: asset-liability ratio (corporate solvency index), value and appreciation rate of capital (corporate development ability index) and R&D expenditure rate (corporate stable operation index). However, the company-level characteristics are not limited to this, but also cover many aspects, such as the company's establishment years, policy dividend level, policy premiums per unit, the company's business procedures rate and so on. Due to the feasibility of the empirical model selected, this paper does not discuss these aspects, and these factors are rarely involved in existing literature, so it can be the future research direction of scholars at home and abroad. This paper considers that these variable data can be obtained through the annual reports disclosed publicly by companies or through field visits to companies, and accordingly, appropriate empirical methods and models are selected to carry out relevant further research.

## Supporting information

**S1 Data.**
(XLS)

**S1 File.**
(DOC)

## Author Contributions

**Conceptualization:** Haisong Dong.

**Data curation:** Haisong Dong.

**Formal analysis:** Ling Tian, Haisong Dong.

**Funding acquisition:** Ling Tian.

**Methodology:** Ling Tian, Haisong Dong.

**Software:** Haisong Dong.

**Supervision:** Ling Tian.

**Validation:** Haisong Dong.

**Writing – original draft:** Haisong Dong.

**Writing – review & editing:** Ling Tian, Haisong Dong.

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
