## [Decision Letter · Decision Letter 0]

1 Sep 2023

PONE-D-23-17007A Firm-level Analysis of Chinese Commercial Health Insurance SurrenderPLOS ONE

Dear Dr. Dong,

Thank you for submitting your manuscript to PLOS ONE. After careful consideration, we feel that it has merit but does not fully meet PLOS ONE’s publication criteria as it currently stands. Therefore, we invite you to submit a revised version of the manuscript that addresses the points raised during the review process.

We look forward to receiving your revised manuscript.

Kind regards,

Eric Fosu Oteng-Abayie

Academic Editor

PLOS ONE

Journal Requirements:

Additional Editor Comments:

Review of the paper is complete. The reviewers suggest major revisions. We have carefully considered the comments and invite you to revise the manuscript to fit the publication requirements of the journal.

Reviewers' comments:

Reviewer's Responses to Questions

**Comments to the Author**

1. Is the manuscript technically sound, and do the data support the conclusions?

Reviewer #1: Partly

Reviewer #2: Yes

Reviewer #3: Yes

2. Has the statistical analysis been performed appropriately and rigorously? 

Reviewer #1: Yes

Reviewer #2: Yes

Reviewer #3: Yes

3. Have the authors made all data underlying the findings in their manuscript fully available?

Reviewer #1: No

Reviewer #2: No

Reviewer #3: Yes

4. Is the manuscript presented in an intelligible fashion and written in standard English?

Reviewer #1: Yes

Reviewer #2: Yes

Reviewer #3: Yes

5. Review Comments to the Author

Reviewer #1: The paper provides valuable insights into the determinants of commercial health insurance surrender in China, utilizing data from seven health insurance companies over an eleven-year period. However, I would like to offer some constructive feedback on the organization of the paper, specifically regarding the introduction, study design, and empirical analysis sections. While the literature review is well-developed and articulated, there are some issues with conceptual clarity in the aforementioned sections. I would like to highlight the following concerns:

1. The authors mention that there are 168 commercial health insurance companies in China as of 2020, but the study only includes data from seven companies. It would be helpful if the authors could provide an explanation for this selection of only seven companies.

2. The introduction section seems to include descriptive statistics, which would be better placed in the empirical analysis section. The introduction should focus on providing background information and addressing the research gap the study aims to fill. Table 1 should be moved to the empirical analysis section, and a condensed version of Section 3 (Theoretical Analysis and Research Hypothesis) should be integrated into the introduction section.

3. The statements in lines 192 to 205 appear to be the authors' own conjectures without proper citations. It is essential to support these statements with relevant theoretical and empirical literature.

4. The study design section lacks comprehensive explanations. The selected panel data econometric model, the fixed effect least square dummy variables (LSDV) model, should be described in detail in the study design section. It would be helpful to clarify why this specific model was chosen over other panel data models and provide information on the statistical (econometric) tests to be conducted and the reasoning behind them. Additionally, robustness checks should be discussed in the study design section, rather than in the empirical results section.

5. Table 3 should be relocated to the empirical results section for better coherence.

6. The results of the various statistical tests (Tables 4, 5, and 6) should be placed in the appendix or supplementary material section, rather than within the main body of the paper.

Taking these suggestions into consideration will enhance the clarity and structure of the paper.

Reviewer #2: Comment 1

The introduction is generally fine in it's current state. However, the motivation for this study is completely missing. The contribution of the study is poorly stated and disorganized. It is utterly confusing the purpose of the literature mentioned. The authors should give a clear distinction between this study and existing studies, and provide a clear literature gap.

Comment 2

The introduction section also did not touch on the main issues in the topic. The authors should conceptualize the terms in the study well and also bring out what other studies have done in this area and what this study seeks to add.

Comment 3

In the literature review section, the authors should first explain the meaning of commercial health insurance surrender and critically evaluate the literature: Instead of simply summarizing the findings of each study. Discuss any inconsistencies in the existing literature and highlight areas where further research is needed.

Comment 4

The authors should change the interpretation of results presented in line 327 to 336. Regression and correlation are not the same. Regression shows an effect of one variable on the other and correlation shows linear association.

Comment 5

The authors should discuss the significance of the coefficients in the regression analysis. It would be useful to provide some economic or practical significance to these findings. Additionally, the authors could compare the findings to existing literature or theoretical expectations to provide a more comprehensive analysis.

Comment 6

The conclusion of the study is weak, the authors need to indicate the major factors that must be improved to reduce surrender rate in insurance sector in China.

Reviewer #3: The following are suggestions to improve the quality of the work

* There is the need to offer justification for focusing on health insurance surrender in China. What is the current situation? what is/are the likely implications if the trend remains same? What are the related policy issues that warrant this study.

* The discussion of the results should be improved. For each KEY variable tell the effect it has on "surrender" and offer possible reasons for the outcome in China. Relate the discussion to the literature and previous empirical studies/findings.

6. PLOS authors have the option to publish the peer review history of their article (what does this mean?). If published, this will include your full peer review and any attached files.

Reviewer #1: **Yes: **Kwadwo Arhin

Reviewer #2: No

Reviewer #3: No

---

## [Author Response · Author response to Decision Letter 0]

27 Sep 2023

Response to Reviewers

Reviewer #1: 

1. The authors mention that there are 168 commercial health insurance companies in China as of 2020, but the study only includes data from seven companies. It would be helpful if the authors could provide an explanation for this selection of only seven companies.

Response：

Thank you very much for your comments. Based on your suggestions, we have explained this accordingly. See the red brackets in "1.Introduction" for specific modifications.

Thanks again for the reviewer's suggestions.

2. The introduction section seems to include descriptive statistics, which would be better placed in the empirical analysis section. The introduction should focus on providing background information and addressing the research gap the study aims to fill. Table 1 should be moved to the empirical analysis section, and a condensed version of Section 3 (Theoretical Analysis and Research Hypothesis) should be integrated into the introduction section.

Response：

Thank you very much for your comments. Based on your suggestions, we have expanded and modified "1 Introduction", for details, see the red part of "1.Introduction". However, we choose to leave Table 1 in the "Introduction", because Table 1 is a brief introduction to each professional health insurance company, so that readers can understand the basic information of each professional health insurance company in China more quickly. Therefore, please kindly understand our approach..

Thanks again for the reviewer's suggestions.

3. The statements in lines 192 to 205 appear to be the authors' own conjectures without proper citations. It is essential to support these statements with relevant theoretical and empirical literature.

Response：

Thank you very much for your comments. Based on your suggestion, we have added relevant quotations to lines 192 to 205, mainly adding the literature that asset-liability ratio has a significant positive impact on surrender rate, because there have been empirical studies on this and reached the same conclusion as this paper. In addition, the significant impact of R&D expenditure rate and capital preservation and appreciation rate on surrender rate, It is the extended research finding of this paper to verify the "policy replacement hypothesis" (at the company level), which is the innovation point of this paper, and this part is the theoretical analysis part of our paper. For details, see lines 192 to 205.

In addition, we attach the relevant literature cited for reference by the reviewers. Attached:

Yu L, Cheng J, Lin T T. Life Insurance Lapse Behaviour: Evidence from China. Geneva Papers on Risk and Insurance-Issues and Practice. 2019, 44(4):653-678. https://doi.org/10.1057/ s41288-018-0104-5

Thanks again for the reviewer's suggestions.

4. The study design section lacks comprehensive explanations. The selected panel data econometric model, the fixed effect least square dummy variables (LSDV) model, should be described in detail in the study design section. It would be helpful to clarify why this specific model was chosen over other panel data models and provide information on the statistical (econometric) tests to be conducted and the reasoning behind them. Additionally, robustness checks should be discussed in the study design section, rather than in the empirical results section.

Response：

Thank you very much for your comments. First, in accordance with your suggestions, we separate the Robustness test and Endogenous statement from the section on "Empirical Analysis" and make them a separate section, with modifications in "6. Robustness test" and "7. Endogenous description".

In addition, we divided three small parts in "4 Research Design", and the third part may cause misunderstanding. Therefore, we changed "4.3 Model setting"to "4.3 Model expression", that is, we modified the part title. Section 4.3 is intended to introduce the model expression. In the model expression, we explain why we are adding a “time trend term” instead of a "time dummy variable" in the usual sense. As for what kind of estimation method is adopted for "long panel data with variable intercept statically unbalanced", through a series of tests in the "5 Empirical Analysis" chapter we concluded that "PW+PCSE" estimation method is the most appropriate estimation method, rather than "LSDV" estimation method.

Thanks again for the reviewer's suggestions.

5. Table 3 should be relocated to the empirical results section for better coherence.

Response：

Thank you very much for your comments. Based on your suggestions, we change "4.2 Data sources and descriptions" to "4.2 Data sources", that is, the title is modified, descriptive statistics of variable data are no longer carried out. Table 3 is repositioned in “5.2 Stability test and descriptive statistics of variable data “”in “5. Empirical Analysis”. For details, see the red section.

Thanks again for the reviewer's suggestions.

6. The results of the various statistical tests (Tables 4, 5, and 6) should be placed in the appendix or supplementary material section, rather than within the main body of the paper.

Response：

Thank you very much for your comments. Since this paper adopts "long panel data with variable intercept statically unbalanced", it is necessary to conduct "multicollinearity test" and "stationarity test". To select a suitable estimation method, therefore, "Test results of inter-group heteroscedasticity, intra-group autocorrelation and inter-group correlation" must be carried out. Therefore, these statistical tests are necessary. At the same time, considering that the length of this paper is not very long, we choose to put them in the body. Therefore, please kindly understand our approach.

Thanks again for the reviewer's suggestions.

Reviewer #2: 

Comment 1: The introduction is generally fine in it's current state. However, the motivation for this study is completely missing. The contribution of the study is poorly stated and disorganized. It is utterly confusing the purpose of the literature mentioned. The authors should give a clear distinction between this study and existing studies, and provide a clear literature gap.

Response：

Thank you very much for your comments. Based on your suggestions, we have made supplementary revisions to the research contributions in this paper, for details, see the contribution section of "2. Review of Relevant Literature".

Thanks again for the reviewer's suggestions.

Comment 2: The introduction section also did not touch on the main issues in the topic. The authors should conceptualize the terms in the study well and also bring out what other studies have done in this area and what this study seeks to add.

Response：

Thank you very much for your comments. Based on your suggestions, we have expanded and revised the Introduction, for details, see the red part of "1. Introduction".

Thanks again for the reviewer's suggestions.

Comment 3: In the literature review section, the authors should first explain the meaning of commercial health insurance surrender and critically evaluate the literature: Instead of simply summarizing the findings of each study. Discuss any inconsistencies in the existing literature and highlight areas where further research is needed.

Response：

Thank you very much for your comments. Based on your suggestions, we have put the definition and meaning of surrender in the "1. Introduction", highlighted the topic of the article, and revised the "2. Review of Relevant Literature", see the red part of the relevant chapter for specific modifications.

Thanks again for the reviewer's suggestions.

Comment 4: The authors should change the interpretation of results presented in line 327 to 336. Regression and correlation are not the same. Regression shows an effect of one variable on the other and correlation shows linear association.

Response：

Thank you very much for your comments. Based on your suggestions, we have reviewed the relevant data again and believe that the interpretation of the results in lines 327 to 336 is acceptable. The reason is that although regression and correlation are different, they are also related, and correlation analysis is the basis and premise of regression analysis. If there is no correlation, it is meaningless to do regression analysis, that is, regression is based on correlation. Here, we emphasize whether the correlation is positive or negative.

In addition, a lot of literature has been expressed in this way, and we attach the relevant literatures for the review of reviewers. The specific literatures are as follows:

①Wang Min, He Jie, Xu Peng. Ultimate Control Rights and Corporate Fraud: Evidence from China [J]. Emerging Markets Finance and Trade, 2022, 58(4):1206-1213. DOI:10.1080/1540496X.2020.1845647（Page 2）

②Ting Shi, Zang Wenbin, Chen Chen, et al. Income distribution and health: What do we know from Chinese data? [J]. PloS one, 2022, 17(1):e0263008-e0263008. DOI: 10.1371/JOURNAL.PONE.0263008（Page 2）

③Shen Lu, He Guohua, Yan Huan. Research on the Impact of Technological Finance on Financial Stability: Based on the Perspective of High-Quality Economic Growth [J]. Complexity, 2022. DOI: 10.1155/2022/2552520（Page 3）

④Wang Qian, Wang Jun, Gao Feng. Who is more important, parents or children? Economic and environmental factors and health insurance purchase [J]. The North American Journal of Economics and Finance, 2021, 58:101479.DOI: 10.1016/J.NAJEF.2021.101479（Page 2）

However, based on your suggestions, we have made corresponding modifications, specifically, see the red part of "2. Review of Relevant Literature" and the red part of "5.4 Analysis of the empirical results".

Thanks again for the reviewer's suggestions.

Comment 5: The authors should discuss the significance of the coefficients in the regression analysis. It would be useful to provide some economic or practical significance to these findings. Additionally, the authors could compare the findings to existing literature or theoretical expectations to provide a more comprehensive analysis.

Response：

Thank you very much for your comments. Based on your suggestions, we discussed the significance of each coefficient in the regression analysis and compared the research results with existing literatures. For details, see the red part of "5.4 Analysis of the empirical results".

Thanks again for the reviewer's suggestions.

Comment 6: The conclusion of the study is weak, the authors need to indicate the major factors that must be improved to reduce surrender rate in insurance sector in China.

Response：

Thank you very much for your comments. Based on your suggestions, we have given emphasis in the Policy Revelation, as amended in red in "8.2 Policy enlightenments".

Thanks again for the reviewer's suggestions.

Reviewer #3: 

1* There is the need to offer justification for focusing on health insurance surrender in China. What is the current situation? what is/are the likely implications if the trend remains same? What are the related policy issues that warrant this study.

Response：

Thank you very much for your comments. Based on your suggestions, we have revised the "Introduction", for details, see the red part of "1. Introduction".

Thanks again for the reviewer's comments.

2* The discussion of the results should be improved. For each KEY variable tell the effect it has on "surrender" and offer possible reasons for the outcome in China. Relate the discussion to the literature and previous empirical studies/findings.

Response：

Thank you very much for your comments. Based on your suggestions, we have revised this, and see the red section of "5.4 Analysis of the empirical results" for details.

Thanks again for the reviewer's comments.

---

## [Decision Letter · Decision Letter 1]

10 Nov 2023

PONE-D-23-17007R1A Firm-level Analysis of Chinese Commercial Health Insurance SurrenderPLOS ONE

Dear Dr. Dong,

Thank you for submitting your manuscript to PLOS ONE. After careful consideration, we feel that it has merit but does not fully meet PLOS ONE’s publication criteria as it currently stands. Therefore, we invite you to submit a revised version of the manuscript that addresses the points raised during the review process.

We look forward to receiving your revised manuscript.

Kind regards,

Eric Fosu Oteng-Abayie

Academic Editor

PLOS ONE

Journal Requirements:

Additional Editor Comments:

Please address the minor concern of Reviewer 2.

Reviewers' comments:

Reviewer's Responses to Questions

**Comments to the Author**

1. If the authors have adequately addressed your comments raised in a previous round of review and you feel that this manuscript is now acceptable for publication, you may indicate that here to bypass the “Comments to the Author” section, enter your conflict of interest statement in the “Confidential to Editor” section, and submit your "Accept" recommendation.

Reviewer #1: (No Response)

Reviewer #2: All comments have been addressed

2. Is the manuscript technically sound, and do the data support the conclusions?

Reviewer #1: Yes

Reviewer #2: Yes

3. Has the statistical analysis been performed appropriately and rigorously? 

Reviewer #1: Yes

Reviewer #2: Yes

4. Have the authors made all data underlying the findings in their manuscript fully available?

Reviewer #1: Yes

Reviewer #2: No

5. Is the manuscript presented in an intelligible fashion and written in standard English?

Reviewer #1: Yes

Reviewer #2: Yes

6. Review Comments to the Author

Reviewer #1: Thank you for your responses to my comments. The subject matter of this article is very important. However, to improve the quality of the paper, I reiterate one of my earlier comments below.

1. The introduction should focus on providing background information and addressing the research gap the study aims to fill. Table 1 should be moved to the empirical analysis section, and a condensed version of Section 3 (Theoretical Analysis and Research Hypothesis) should be integrated into the introduction section.

Reviewer #2: (No Response)

7. PLOS authors have the option to publish the peer review history of their article (what does this mean?). If published, this will include your full peer review and any attached files.

Reviewer #1: **Yes: **Kwadwo Arhin

Reviewer #2: **Yes: **Gideon Mensah

---

## [Author Response · Author response to Decision Letter 1]

11 Nov 2023

Response to Reviewers

Reviewer #1: 

Thank you for your responses to my comments. The subject matter of this article is very important. However, to improve the quality of the paper, I reiterate one of my earlier comments below.

1. The introduction should focus on providing background information and addressing the research gap the study aims to fill. Table 1 should be moved to the empirical analysis section, and a condensed version of Section 3 (Theoretical Analysis and Research Hypothesis) should be integrated into the introduction section.

Response：

Dear Reviewer Kwadwo Arhin:

Thank you very much for your comments. Based on your suggestion again, we have revised this, specifically see the red part of "1. Question Raised and Research Hypothesis".

Thanks again for your suggestion.

Reviewer #2: 

4. Have the authors made all data underlying the findings in their manuscript fully available?

Response：

Dear Reviewer Gideon Mensah:

Thank you very much for your comments. Based on your suggestion, We have deleted the subdata and provided the master data, as detailed in “Data.xls”.

Thanks again for your suggestion.

---

## [Decision Letter · Decision Letter 2]

18 Dec 2023

A Firm-level Analysis of Chinese Commercial Health Insurance Surrender

PONE-D-23-17007R2

Dear Dr. Dong,

We’re pleased to inform you that your manuscript has been judged scientifically suitable for publication and will be formally accepted for publication once it meets all outstanding technical requirements.

Kind regards,

Eric Fosu Oteng-Abayie

Academic Editor

PLOS ONE

Additional Editor Comments (optional):

Reviewers' comments:

Reviewer's Responses to Questions

**Comments to the Author**

1. If the authors have adequately addressed your comments raised in a previous round of review and you feel that this manuscript is now acceptable for publication, you may indicate that here to bypass the “Comments to the Author” section, enter your conflict of interest statement in the “Confidential to Editor” section, and submit your "Accept" recommendation.

Reviewer #1: All comments have been addressed

Reviewer #2: All comments have been addressed

2. Is the manuscript technically sound, and do the data support the conclusions?

Reviewer #1: Yes

Reviewer #2: Yes

3. Has the statistical analysis been performed appropriately and rigorously? 

Reviewer #1: Yes

Reviewer #2: Yes

4. Have the authors made all data underlying the findings in their manuscript fully available?

Reviewer #1: Yes

Reviewer #2: Yes

5. Is the manuscript presented in an intelligible fashion and written in standard English?

Reviewer #1: Yes

Reviewer #2: Yes

6. Review Comments to the Author

Reviewer #1: The title of section 1 should be changed from "Question Raised and Research Hypothesis" to "Introduction".

Reviewer #2: (No Response)

7. PLOS authors have the option to publish the peer review history of their article (what does this mean?). If published, this will include your full peer review and any attached files.

Reviewer #1: **Yes: **Kwadwo Arhin

Reviewer #2: **Yes: **Gideon Mensah

---

## [Editor Report · Acceptance letter]

28 Dec 2023

PONE-D-23-17007R2 

PLOS ONE

Dear Dr. Dong, 

I'm pleased to inform you that your manuscript has been deemed suitable for publication in PLOS ONE. Congratulations! Your manuscript is now being handed over to our production team.

Kind regards, 

on behalf of

Dr. Eric Fosu Oteng-Abayie 

Academic Editor

PLOS ONE